# Mechanical Assessment of Fatigue Characteristics between Single- and Multi-Directional Cyclic Loading Modes on a Dental Implant System

**DOI:** 10.3390/ma13071545

**Published:** 2020-03-27

**Authors:** Won Hyeon Kim, Eun Sung Song, Kyung Won Ju, Dohyung Lim, Dong-Wook Han, Tae-Gon Jung, Yong-Hoon Jeong, Jong-Ho Lee, Bongju Kim

**Affiliations:** 1Clinical Translational Research Center for Dental Science, Seoul National University Dental Hospital, Seoul 03080, Korea; wonhyun79@gmail.com (W.H.K.); songeunsung@gmail.com (E.S.S.); 2Department of Mechanical Engineering, Sejong University, Seoul 05006, Korea; dli349@sejong.ac.kr; 3Department of Oral and Maxillofacial Surgery and Dental Research Institute, School of Dentistry, Seoul National University, Seoul 03080, Korea; kwcindyju@hotmail.com; 4Department of Cogno-Mechatronics Engineering, College of Nanoscience & Nanotechnology, Pusan National University, Busan 46241, Korea; nanohan@pusan.ac.kr; 5Department of Medical Device Development Center, Osong Medical Innovation Foundation, Chungbuk 28160, Korea; bygon@kbiohealth.kr (T.-G.J.); yonghoonj186@kbiohealth.kr (Y.-H.J.)

**Keywords:** multi-directional loadings, fatigue, vertical fracture, dental implant, worst case, mechanical testing

## Abstract

Mechanical testing based on ISO 14801 standard is generally used to evaluate the performance of the dental implant system according to material and design changes. However, the test method is difficult to reflect on the clinical environment because the ISO 14801 standard does not take into account the various loads from different directions during chewing motion. In addition, the fracture pattern of the implant system can occur both in the horizontal and the vertical directions. Therefore, the purpose of this study was to compare fatigue characteristics and fracture patterns between single directional loading conditions based on the ISO 14801 standard and multi-directional loading condition. Firstly, the static test was performed on five specimens to derive the fatigue load, and the fatigue load was chosen as 40% of the maximum load measured in the static test. Subsequently, the fatigue test was performed considering the single axial/occlusal (AO), AO with facial/lingual (AOFL) and AO with mesial/distal (AOMD) directions, and five specimens were used for each fatigue loading modes. In order to analyze the fatigue characteristics, the fatigue cycle at the time of specimen fracture and displacement change of the specimen every 500 cycles were measured. Field emission scanning electron microscopy (FE-SEM) was used to analyze the fracture patterns and the fracture surface. Compared to the AO group, the fatigue cycle of the AOFL and AOMD groups showed lower about five times, while the displacement gradually increased with every 500 cycles. From FE-SEM results, there were no different surface morphology characteristics among three groups. However, the AOMD group showed a vertical slip band. Therefore, our results suggest that the multi-directional loading mode under the worst-case environment can reproduce the vertical fracture pattern in the clinical situation and may be essential to reflect on the dental implant design including connection types and surface treatments.

## 1. Introduction

Dental implants were widely used to treat tooth defects caused by tooth decay, periodontitis and tooth damage [1,2,3]. Dental implants have been constantly developing including the connection between the abutment and crown advancing from external type connection to currently used internal type to overcome the shortcomings of external type of implants developed by the previous study [4]. In the case of the external type, Branemark’s protocol indicated that the complete restoration of the edentulous arch can be done through torque transfer and coupling mechanism [4]. However, due to a wider range of applications of dental implants nowadays, limited height and joint instability due to insufficient contact between the fixture and abutment and excessive bending load from off-axis load have led to implant fracture and frequent abutment screw loosening [5,6,7,8,9]. In order to overcome this problem, advancements in the modification of the contact area of abutment and fixture have brought us the internal type of dental implants. Due to the wide contact area of abutment and fixture in the internal type, resistance to bending load has increased [10,11]. Moreover, resistance to rotation and abutment screw loosening have decreased in the morse tapered type due to friction fit structure between the fixture and abutment interface and cold welding mechanism [12]. However, various other factors such as screw loosening of the abutment on the fixture, breakage of the prosthesis, fracture of abutment screw, and peri-implantitis leading to implant failure have also been reported [13,14,15]. To improve the structures and designs of dental implants, researches have not only focus on the modification of the material or implant design but also on the mechanism of occlusal forces, stress analysis [16,17] and shear-compression fatigue test following the ISO 14801 standard [18] have been actively carried out internationally. Among these methods, a mechanical test based on the ISO 14801 standard is useful for comparing implants of different designs, sizes and assembled states. This test indicates the pattern of structural failures in the implant system. However, the ISO 14801 standard [18] has its limitation of only considering the vertical load condition which did not take into account the loads from different directions during chewing motion. In addition, the experimental jig used in ISO 14801 standard has a round shape in the form of the holder, but since the teeth in the actual oral cavity have a concave occlusal surface, the direction of the load, load path and pattern may be generated differently at the contact surface. Food between teeth can cause two phases: fast closure and slow closure [19,20]. A previous study set the grinding phase at the occlusal surface, which is similar to the load path and pattern generated in the actual teeth, to implement mechanisms such as vertical loading at the occlusal surface and movement along the surface [21]. A previous study also showed the actual chewing cycle as range 1–1.58 Hz [22,23], unlike how the fatigue load cycle is specified as 15 Hz in the ISO 14801 standard. In addition, chewing movement has a complex range of directional loads, not only limited to vertical loading but also oblique and horizontal loading which are guided by the temporomandibular joint moves with up to 6 degrees of freedom in cartesian space [24].

The internal connection type of dental implants showed vertical fracture on the transverse plane of the implant vertical to fracture line when clinical or physiological occluding force was applied [25]. Whereas, a previous literature reported horizontal fracture on the transverse plane of the implant fracture line parallel to fixture platform during the implant fatigue test of the internal connection type implants following the ISO 14801 standard [26]. For this reason, the single-directional loading condition was different from the masticatory loading condition, and it was difficult to reproduce the damage, fatigue characteristics of the dental implant in the oral cavity. Therefore, it needs to develop a cyclic loading experiment protocol simulating the considered oral cavity and masticatory loading.

The testing hypotheses were as follows: (1) worst-case situation will occur under multi-directional load conditions rather than single load condition; and (2) fixation failure with vertical fracture pattern on the fixture under multi-directional loading condition will occur. Therefore, this study aims to compare the fatigue characteristics and fracture patterns between single directional loading condition and multi-directional loading condition with facial/lingual (FL) and mesial/distal (MD) directions.

## 2. Materials and Methods

### 2.1. Specimen Preparation

A total of 20 dental implants including an abutment, fixture, and abutment screw made with Ti-6Al-4V alloy material (Osstem Implant Co. Ltd., Seoul, Korea) were used in this mechanical study (Figure 1). All implant specimens consist of an internal connection between abutment and fixture. All fixtures were 10 mm in length, 4.5 mm in diameter and with 0.8 mm screw thread pitch. Twenty implant specimens were divided into two different kinds of mechanical test; five for static test and fifteen for dynamic fatigue tests which were further divided into three groups (five for the AO group, AOFL group and AOMD group). The abutment was connected to the fixture with an abutment screw using the recommended torque of 30 Ncm following the manufacturer’s instruction.

### 2.2. Mechanical Test Preparation: Fixation of Specimens

An MTS 858 Material Testing Machine (MTS Bionix Tabletop Systems, MTS systems corp., Eden Prairie, MN, USA) was used for static shear-compression test through monotonic loading. On the other hand, a joint simulator (ADL-Force 5, Advanced Mechanical Technology Inc., Watertown, MA, USA) was used for a dynamic shear-compression fatigue test.

Generally, all mechanical test setup of the dental implant system was performed according to the ISO 14801:2016 standard with a polished flat applicator that permitted free movement of a hemispherical loading member when loaded (Figure 2). For a static test, the central longitudinal axis (Line DE) of the implant-abutment was performed at a 30° angle to the loading direction (Line AB) of the testing machine (Figure 2). The distance between the intersection of the line DE and line AB and the fixation region of the fixture was set to 11 mm. The distal portion of the fixture was fixed with a collet chuck holder 3 mm away from the platform, mimicking in relative to having a 3 mm marginal bone loss after surgery as reported [27].

### 2.3. Mechanical Test Methods: Static and Dynamic Shear-Compression Tests

A static test was performed with the vertical load at a speed of 1 mm/min until the failure of the dental implant systems [28]. Five specimens were used to test for static shear-compression fatigue to determine the optimal testing load to be applied in dynamic cyclic loading. Failure of the static test was defined as having a fracture in any of the implant system (abutment, fixture or abutment screw), and/or sudden drop of load resistance observed through the load-displacement curve [28,29]. Subsequently, the dynamic fatigue test will be performed with 40% value of the maximum load given by the static test and 5 Hz loading frequency. The value of 40% of the maximum load derived from the static test in our study considered the maximum bite force mentioned in the literature [30]. The average value of the right and left molar bite forces of male and female was mentioned in as 577 N, which was similar to the derived value (570 N) [30]. The biting frequency was reported to be 1–1.58 Hz, but this study set the value as 5 Hz to confirm the fracture frequency and pattern between axial and multi-directional load to consider the worst biting frequency. Teeth mechanical tests in most studies used 5 Hz [23,31,32,33,34]. Dynamic shear-compression tests were performed in three different directions. A single direction was performed with only axial directional force application for the AO group (Figure 3a), while multi-directional was performed with a 2° angulation for AOFL and AOMD groups (Figure 3b,c, respectively) assuming human chewing-like environment. Because the occlusal surface of teeth is unbalanced when chewing food or during mastication, loading is generated at various inclination angles from 2° to 34° in the previous study [35]. Therefore, in order to analyze the effect on fatigue fracture when an additional inclined load was generated on another plane at the single load, a rotation condition of ±2° on the facial-lingual and mesial-distal planes was added. The displacement was measured every 500 cycles in order to compare the displacement of implants according to different loading directions. Failure of the shear-compression test was defined as having excessive displacement of the implant system, reduction of the sinusoidal load observed at the cyclic-load waveform, and/or presence of fracture line in one of the implant systems.

### 2.4. Surface Morphological Analysis

After fatigue tests, all the test specimens were analyzed for the damaged surface and the crack propagation. The fracture shapes were analyzed with a digital picture for the macro image. Thereafter, the surfaces of the fractured specimen were identified with field emission scanning electron microscope (FE-SEM) (Hitachi SU8230, Tokyo, Japan) set at a 15 kV acceleration voltage for micro-structure to determine the crack propagation and direction. Fractured micro surface images were obtained with magnification between ×30 and ×3k, and evaluated the surfaces whether it has a dimple, facet layer, or slip band according to fatigue loading.

### 2.5. Statistical Analysis

The failure cycles of the fatigue test between three groups were presented as mean ± standard deviation (S.D.), and one-way analysis of variance and a Tukey’s honestly significant difference (HSD) post-hoc test were chosen to assess differences among the loading modes concerning the failure cycle and displacement at failure during a fatigue test. The level of significance was set at *p*-value < 0.05. Statistical analysis was performed using SigmaPlot (14.0, Systat Software Inc., San Jose, CA, USA).

## 3. Results

### 3.1. The Result of the Static Shear-Compression Test

The mean values of the maximum load and displacement at failure for the static test were presented in Table 1. The maximum load at the static shear-compression test showed an average of 1423 N and a standard deviation of 81 N. The displacement at failure showed an average of 1.75 mm and a standard deviation of 0.31 mm.

The dynamic fatigue test was performed with a 40% value of the average maximum load from the static test (1423 N) which is 570 N and 5 Hz loading frequency. The cyclic load was set to vary sinusoidally with a 40% value of the maximum load given by the static test which is 570 N and 10% of this value which is 57 N showing the fatigue waveform for single directional loading (Figure 4a). For the multi-directional loading condition, ±2° was additionally applied at each facial–lingual and mesial–distal directions on the axial fatigue load in single directional loading (Figure 4b).

### 3.2. Failure Cycles and Displacement Change of Fatigue Test

Failure cycles of fatigue test were measured as 22,008 ± 6,332 in the AO group, 4,731 ± 806 in the AOFL group, and 4297 ± 509 in the AOMD group. AOFL and AOMD groups were found to have approximately five times less than the AO group (Figure 5a). The failure cycles in the AO group was statistically higher than the AOFL group and the AOMD group (*p* < 0.001 and *p* < 0.001, respectively). However, the differences on fatigue failure cycles between the AOFL group and the AOMD group were not statistically significant (*p* = 0.981).

In addition, to evaluate the effects of single- and multi-directional loadings, the deformation of specimens was measured every 500 cycles in the fatigue test. In all cycles, the AO group showed statistically less deformation than the AOFL and the AOMD groups (*p* < 0.001 and *p* < 0.001, respectively). On the other hand, there was no statistical difference between the AOFL and the AOMD groups. For every 500 cycles from 1111 cycles to 3111 cycles, the *p*-values were 0.840, 0.877, 0.864, 0.882 and 0.883, respectively. For AOMD and AOFL groups which were under the multi-directional loading condition, the axial displacement increased every 500 cycles from 1111 cycles, but in the AO group which is under single loading condition, there was hardly any increase from 1111 to 3111 cycles (Figure 5b).

Figure 6 showed a fractured implant system after fatigue tests for three loading conditions. In this mechanical test, it is possible to observe the horizontal fracture of the fixture and/or the abutment screw in the AO group, while the vertical fracture of the fixture in the AOFL and the AOMD groups.

### 3.3. Fracture Pattern of the Implants

As based on the results from FE-SEM micrographs, there were no different surface morphology characteristics between three groups for single and multi-directional loading (Figure 7). However, in the direction of the slip band from a comparison of the AO and the AOMD groups, only the AOMD group showed a vertical slip band with crack propagation direction (Figure 7c). For the SEM image magnified 1000 times, the secondary fracture was generated at the final fracture site for all three groups, but at the crack initiation site, the secondary fracture was generated on the specimen that was applied multi-directional loading (Figure 7).

## 4. Discussion

Based on the existing ISO standard, the fatigue characteristics test according to single directional load takes into account the worst case of dental implants; it is performed by fixing at 3 mm away from the lower end of the fixture platform, attaching a semi-circular jig at the abutment and applying vertical load [18,26]. This type of vertical loading continuously generates compressive deformation at one side and tensile strain at the opposite side of the fixture near the fixture platform, causing fractures in which the fixture or abutment screw is parallel to the fixture platform. Most of the biomechanical experiments mentioned above that applied single directional loads showed that the fixture and abutment screws fractured parallel to the fixture platform [26,36,37,38].

However, the masticatory loading condition in the oral cavity showed complex loads with various angles are applied to the occlusal surfaces of the teeth by food, unlike single loading conditions [19,20,21,24]. To reproduce the assessment taking into account the oral clinical environment, the movement condition of the occlusal surface and the loading condition at facial-lingual and mesial-distal directions should be considered.

Therefore, to complement the limitations of the widely used performance evaluation of dental implants, this study aimed to reproduce the load path and pattern applied to the tooth in the oral cavity by continuously applying the multi-directional load, and to analyze the fatigue characteristics according to multi-directional loads, the testing method was devised by adding lingual-facial and mesial-distal directions to the single directional load based on the existing ISO 14801.

In the experiment of the AO group of single directional fatigue load based on the ISO standard and the multi-directional AOFL and AOMD groups which increased the degrees of freedom at facial-lingual and mesial-distal directions, the two groups of multi-directional loading condition were found to have 5 times lower fatigue cycle value and showed statistically significant difference. As with the fatigue cycle results, the axial deformation of specimens every 500 cycles showed a similar trend. The AO group showed a constant change by an increase of 500 cycles from 1111 to 3111 fatigue cycles, but the axial displacement of the multi-directional AOFL and AOMD groups increased gradually every 500 cycles (Figure 6b). At the fracture pattern of each specimen after fatigue failure in the AO group, the fixture or the screw was fractured parallel to the fixture platform which is similar to the existing studies using ISO 14801 standard [26,28,38,39]. The fatigue fracture cycles using ISO 14801 standard in previous studies showed approximately 200,000 cycles at 420 N fatigue load and approximately 10,000 cycles at 410 N [28,39]. The fatigue fracture cycle in this study was 22,008 cycle, which was different compared to the previous studies [28,40]. Because the implant design, clamping type, and the fatigue load applied in the fatigue experiment was different, the fatigue fracture cycle derived in this study is considered different from previous studies. Whereas, in the multi-directional loading condition, the fracture at the initial section was parallel to the platform but it showed a vertical fracture at the fracture end heading to the top of the fixture. A previous study also confirmed the vertical fracture pattern of the fixture in vitro test and in silico analysis [41]. However, the vertical fracture did not occur under our single loading condition because of different fixture types with the previous study. It means that fatigue fracture pattern is affected by design of fixture. The vertical fracture shown in our study seemed similar to the clinical study by Jimbo, R. et al. which mentioned vertical fracture of the fixture [25]. The fracture pattern in this study seemed different to the actual clinical vertical fracture [25], but this may be because the fracture occurred when the fixture is completely adhered to by gum and surrounding bone. On the other hand, the experimental environment assumed marginal bone loss, with excessive bending moment and tensile load at the fixture platform. It is considered that this fracture pattern was shown as the crack occurred parallel with the platform at the initial stage then vertical at the last region of the crack. For this reason, compared to the ISO standard experiment protocol, a multi-directional loading condition with increased degrees of freedom was considered to reproduce the worst environment of inducing early fatigue fracture and was confirmed its possibility of observing vertical fracture pattern which may occur clinically.

Moreover, FE-SEM was used to compare the fatigue fracture surface characteristics by single- and multi-directional loads. In the FE-SEM images, a slip band, facet or dimple were observed in 3 groups but there was no morphological difference. However, there was a difference in direction of the slip band between the AO and the AOMD groups, and the AOMD group showed a vertical slip band with crack propagation direction. Under multi-directional loading conditions, the displacement of the specimen may be increased due to the slip band generated on the surface [42], and this may have induced early fatigue fracture. Therefore, it is expected that when vertical loading is applied on the teeth at various directions, the tensile and compressive forces would also be multi-directional on the implant, which would increase displacement due to slip band formation and induce a vertical fracture pattern.

Our limitation used only SLA (sandblasted, Largegrit, Acid-etched)-treated fixture. However, previous studies reported that fatigue resistance and crack generation are significantly affected by surface treatment [43,44,45]. A dental implant especially improved fatigue resistance because of the layer of compressive residual stress by the shot blasting [43,44]. Therefore, future study is needed to analysis the fracture pattern and resistance according to types of surface treatment under the same condition including design and multi-loading condition.

## 5. Conclusions

The fatigue fracture pattern, fracture shape, and fatigue cycle were compared and analyzed between single directional loading condition and multi-directional loading condition which added inclination angle to the teeth on facial-lingual and mesial-distal planes. Our result showed that multi-directional loading conditions generated lower fracture cycles compared to single-directional loading conditions. Vertical fracture of the fixture, a fractured form often reported in clinical results, occurred under multi-directional loading conditions which added facial-lingual and mesial-distal directions.

Therefore, our results suggest that the multi-directional loading mode under the worst-case environment can reproduce the vertical fracture pattern in the clinical situation and may be essential to reflect on the dental implant design including connection types and surface treatments.

## Figures and Tables

**Figure 1 materials-13-01545-f001:**
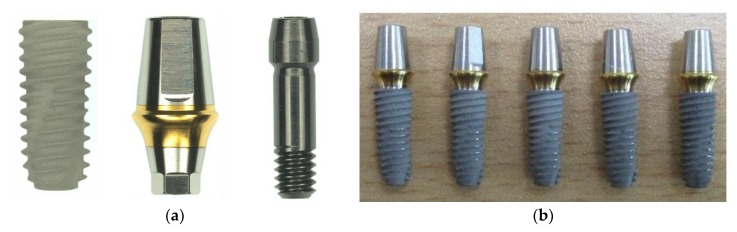
Component of implant systems for static and dynamic experiments. (**a**) Fixture (GSTA5620) with 10 mm of length and 4.5 mm of diameter, the abutment (TS3S5011S) and abutment screw (GSABSS) and (**b**) assembled implant specimens.

**Figure 2 materials-13-01545-f002:**
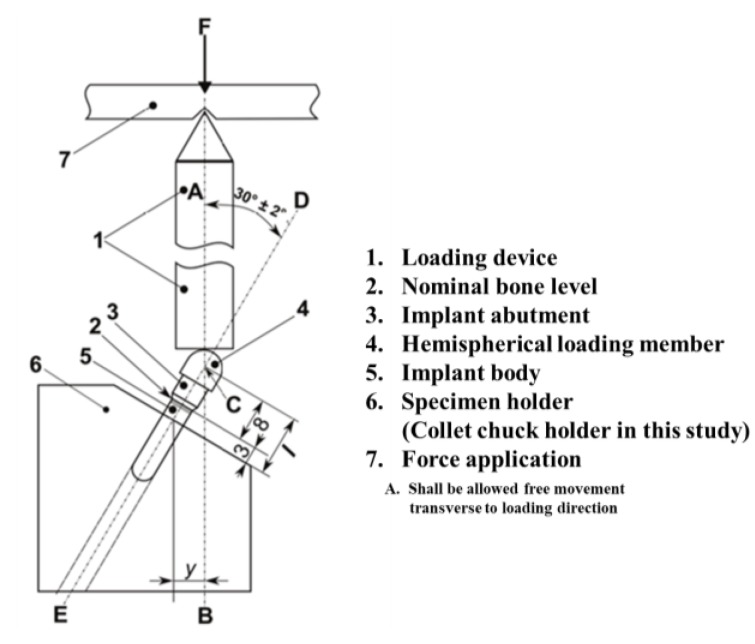
Schematic of test set-up with based on the ISO 14801:2016.

**Figure 3 materials-13-01545-f003:**
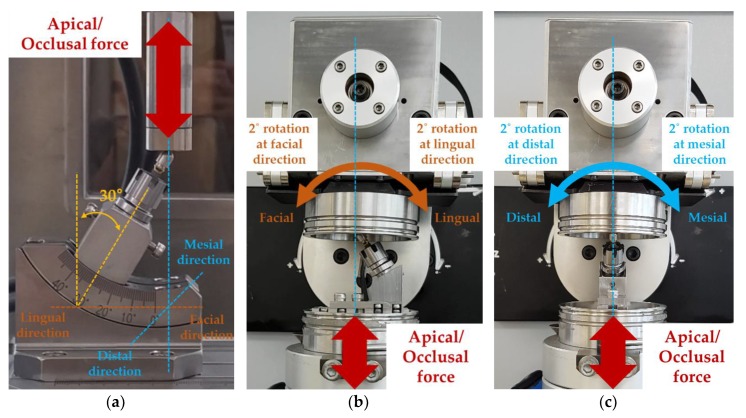
Schematic of test set up for single and multi-directional fatigue test: (**a**) single directional loading (axial/occlusal (AO) group); (**b**) multi-directional loading including facial and lingual directions (AO with facial/lingual (AOFL) group); and (**c**) multi-directional loading including mesial and distal directions (AO with mesial/distal (AOMD) group).

**Figure 4 materials-13-01545-f004:**
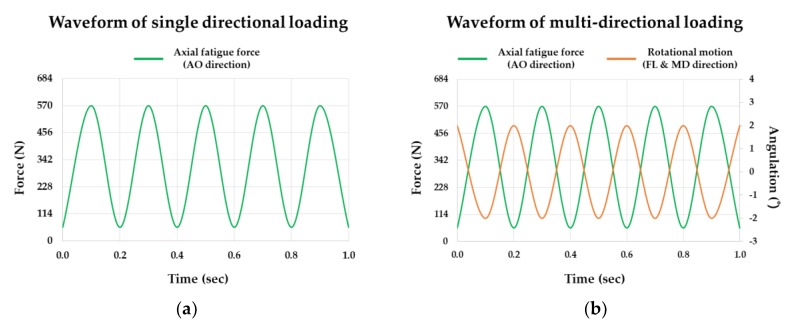
The loading waveform of mechanical test used in this study: (**a**) single directional loading; (**b**) multi-directional loading.

**Figure 5 materials-13-01545-f005:**
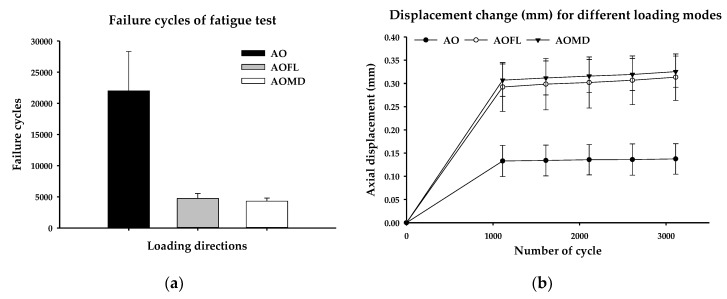
Failure cycles and displacement (mm) measured every 500 cycles with single-directional (AO) and multi-directional (AOFL and AOMD) fatigue loadings: (**a**) failure cycle under different loading conditions; (**b**) displacement (mm) measured every 500 cycles.

**Figure 6 materials-13-01545-f006:**
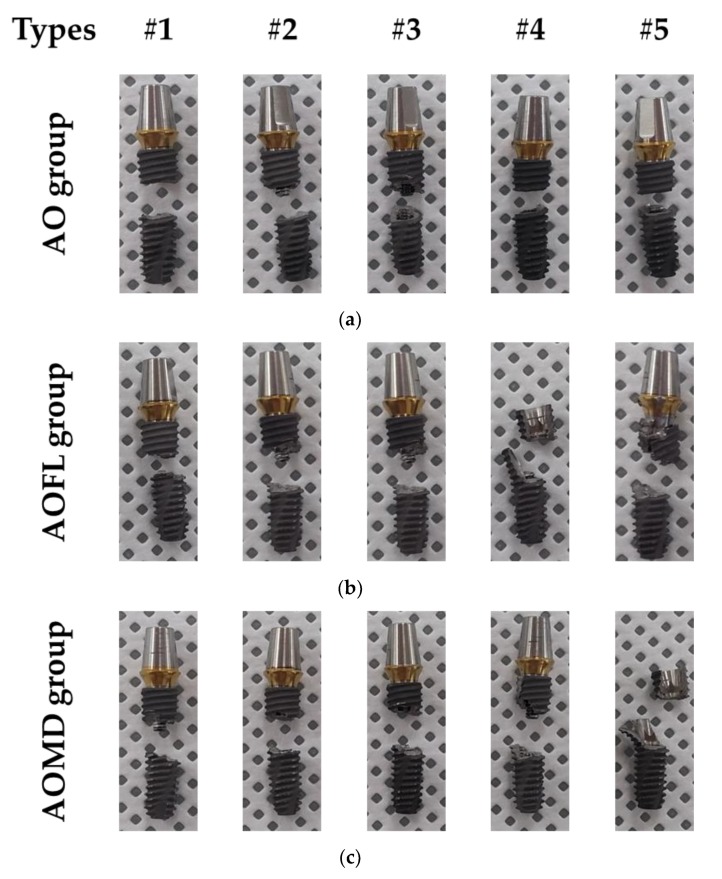
The fracture pattern of specimens after fatigue test through different loading directions: (**a**) single directional loading; (**b**) facial–lingual (FL) multi-directional; and (**c**) mesial–distal (MD) multi-directional loadings.

**Figure 7 materials-13-01545-f007:**
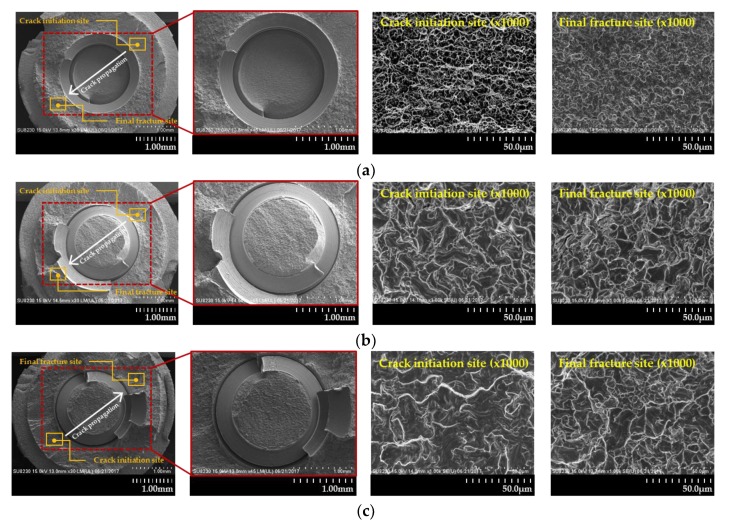
FE-SEM images on the fractured surface of specimens after fatigue test: (**a**) AO group; (**b**) AOFL group and (**c**) AOMD group.

**Table 1 materials-13-01545-t001:** The maximal load for five specimens under static shear-compression test.

Specimens	Maximum Load (N)	Displacement at Failure (mm)
1	1347	1.68
2	1343	2.12
3	1460	1.46
4	1537	1.46
5	1428	2.01
Average	1423	1.75
Standard deviation	81	0.31

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
