# Peer review of "Mechanical Assessment of Fatigue Characteristics between Single- and Multi-Directional Cyclic Loading Modes on a Dental Implant System"

_materials, 2020, doi:10.3390/ma13071545_

Round 1

Reviewer 1 Report

This is an interesting work that fits within the scope of "Materials" journal. The authors proposed a modification of the test method of dental implants. The proposal of using an multi-directional loading seems correct.

I have the following observations:

1) Materials and methods: The descripion of fatigue tests does not explain why a dynamic load of 40% static load and a frequency of 5 Hz were assumed.

2) Results/Discusion: The authors attempted to modify the test method to physiological loading conditions. The number of fatigue cycles to damage is much smaller than those occurring under physiological conditions. Please explain.

3) Results/Discusion: The text does not describe SEM images made at 1000x magnification. The text is missing links to these figures. SEM images scale is illegible.

4) Conclusions: Conclusions are not very insightful.

Generally, the article lacks a thorough fractographic analysis and the strength and fatigue durability of of tested dental implant materials. I suggest developing an analysis and description of these problems. 

Author Response

Dear Reviewer 1

Please confirm the attached file.

Sincerely yours,

Reviewer 2 Report

Dear Authors,

The aim of this study was to evaluate fatigue characteristics and fracture patterns of dental implants under single/multi directional loading conditions. While the topic is fitting to the journal scope, some concerns were raised. Revise the manuscript by following comments.

Major points

The aim described in the Abstract section and the Introduction section were different. Revise them.

Figure 1b should be close up and make arrangement for each component.

Figure 2 was not meaningful and should be removed.

The loading conditions of AOFL and AOMD were not clear. Figure 4b and 4c should be revised. What's the meaning of 2 degree angulation?

Fisher's PLSD test is not recommend to use for avoiding the type 1 error. Why Tukey's poshoc test was not used?

If gold coating was applied on specimens for SEM observation, add the explanation in the section 2.4.

The Discussion section is too short. The following paper might be referred.

Fracture patterns after fatigue testing were observed in same direction as displacement vectors obtained by FEA study. This suggest that the loading direction is related to those fracture patterns.

Yamaguchi S, et al., In vitro fatigue tests and in silico finite element analysis of dental implants with different fixture/abutment joint types using computer-aided design models. J Prosthodont Res 62(1):24-30, 2018.

Minor points

Introduction

"DO GH" should be modified to "Do GH".

"et al" should be described in Italic font. Make sure other related points.

Author Response

Dear Reviewer 2

Please confirm the attached file.

Sincerely yours,

Reviewer 3 Report

The manuscript describe the limits of the standard 14801 ISO for the mechanical assessment of fatigue resistance of dental implants. The author reported that the worst-case environment is represented by multi-directional loading mode, instead of that proposed by the 14801 ISO. The study is well designed and conducted, but some points have to be addressed before its acceptance for publication.

Abstract

  • Do the authors propose to change and update 14801 ISO standard?
  • Revise the English; for example line 34 "showed about five time lower", the sentence is incomplete. Line 36 "only the AOMD group". Line 38-39 "more clinical situation" is not formal.

Introduction

  • The first part of the introduction should be revised because is not clear, a more clear distinction bewteen internal and external conntections must be provided
  • If it is possible provide more references about ISO 14801 limits (line 65-75)
  • Sentences reported in line 60-64 are misleading, it seems that ISO4801 has been already revised in order to consider the multi-directional loads, thus the contribution of this work become unclear; also line 89-90 can be revised in order to make the contribution clearer
  • Line 58, do not cyte a reference with the author's first name
  • Line 71, it is reported that the actual chewing cycle frequency is 1 Hz, so why the authors chose 5 Hz?
  • Line 80-84: The choice of tenses and sentences form must be revised in a more formal way
  • Bibliography is quite old

Materials and methods

  • Can the author provide a code number, or a name, in order to identify the dental implant model used?
  • Do the 20 samples come from the same batch?
  • Was there any indication that 5 samples for each group was the correct sample size? 
  • The method reported in paragraphs 2.3 should be described in a clearer and more rigorous way; moreover, the choice and the variation of the 2° angulation should be motivated clearly
  • line 127-128, state where the choice of 40% of maximum load and 5 Hz of frequency come from 
  • Were the data normally distributed? how was it verified? Was the homoscedasticity verified?

Results

  • in line 162 Table 1 is cited, why?
  • line 163-165, revise the sentence; moreover, Figure 5 should be cited and explained in a proper way 
  • In Figure 5 all the information in the graph are not clearly explained and it is not easy to comprehend angle and load variations for AO and for FL & MD
  • Axes reported in the graph of Figure 6b should start from 0, moreover, the number of cycle should represent the average data of the three groups till failure; maybe in order to make the graph comprehensible, a break can be introduced on the horizontal axis
  • The fractured implants reported in Figure 7 are a representation of all the samples tested in each group? this should be clearly stated. Otherwise a table reporting the type of fractures in each group could be helpful
  • Line 182: what the authors mean with "constant change" with respect to "gradually increase"

Discussion

  • again, please carefully revise the English... sentences like those reported in line 208 or in line 211 are must be revised in a more formal way
  • values of failure cycles should be compared with the literature; recent works which reported results according to ISO 14801 should be considered in the discussion

Conlcusions

  • Revise the English... line 253: "the dental implants tested with the multudirectiona loading... showed... than those tested in accordance... Line 257-258, "a more or less similar clinical..." must be rewritten in a more formal way
  • Line 260 the connection of this statement with the previous one is unclear

Author Response

Dear Reviewer 3

Please confirm the attached file.

Sincerely yours,

Round 2

Reviewer 2 Report

Dear Authors,

The manuscript was well revised by following reviewer's comments.

Author Response

Dear Reviewer

Thank you for your comment.

Sincerely your,